# Synthetic Biology towards Improved Flavonoid Pharmacokinetics

**DOI:** 10.3390/biom11050754

**Published:** 2021-05-18

**Authors:** Moon Sajid, Chaitanya N. Channakesavula, Shane R. Stone, Parwinder Kaur

**Affiliations:** UWA School of Agriculture and Environment, The University of Western Australia, Perth, WA 6009, Australia; moon.sajid@research.uwa.edu.au (M.S.); 22865222@student.uwa.edu.au (C.N.C.); shane.ramsay.stone@gmail.com (S.R.S.)

**Keywords:** flavonoids, pharmacokinetics, anticancer compounds, ex planta synthesis

## Abstract

Flavonoids are a structurally diverse class of natural products that have been found to have a range of beneficial activities in humans. However, the clinical utilisation of these molecules has been limited due to their low solubility, chemical stability, bioavailability and extensive intestinal metabolism in vivo. Recently, the view has been formed that site-specific modification of flavonoids by methylation and/or glycosylation, processes that occur in plants endogenously, can be used to improve and adapt their biophysical and pharmacokinetic properties. The traditional source of flavonoids and their modified forms is from plants and is limited due to the low amounts present in biomass, intrinsic to the nature of secondary metabolite biosynthesis. Access to greater amounts of flavonoids, and understanding of the impact of modifications, requires a rethink in terms of production, more specifically towards the adoption of plant biosynthetic pathways into ex planta synthesis approaches. Advances in synthetic biology and metabolic engineering, aided by protein engineering and machine learning methods, offer attractive and exciting avenues for ex planta flavonoid synthesis. This review seeks to explore the applications of synthetic biology towards the ex planta biosynthesis of flavonoids, and how the natural plant methylation and glycosylation pathways can be harnessed to produce modified flavonoids with more favourable biophysical and pharmacokinetic properties for clinical use. It is envisaged that the development of viable alternative production systems for the synthesis of flavonoids and their methylated and glycosylated forms will help facilitate their greater clinical application.

## 1. Introduction

Flavonoids are one the most abundant and broadly distributed families of biologically active plant natural products (PNP). Chemically, flavonoids have a C6-C3-C6 skeleton, with two phenyl aromatic rings (A and B) along with a heterocyclic ring (C-ring). Based on the substitution of the basic skeleton and B-ring attachment, flavonoids have been split into several subclasses, for example, flavanones, flavones, flavonols, flavan-3-ols, isoflavones, and isoflavanones (Figure 1) [1]. Over 15,000 flavonoids have been identified to date, from many plant families, mainly from legumes.

The role of isoflavones as chemopreventive compounds has been well-established. A number of epidemiological studies, along with retrospective meta-analysis and prospective observational studies, have established that flavonoids possess anticancer activities [2,3,4,5]. As well as these, flavonoids possess a range of pharmacological effects as well as antimicrobial, antioxidant and cardioprotective properties [6]. Structural similarities with estrogen hormone and resulting interactions with cellular signaling cascades make flavonoids an interesting class to be pursued for drug discovery [7].

Several flavonoids have been analysed for their anticancer activities, both in vitro and in vivo, including daidzein, quercetin, silymarin, luteolin, kaempferol, and apigenin [8]. These compounds are active against prostate, colorectal, breast, thyroid, lung and ovarian cancer, including others [9]. Flavonoids mediate their function in multiple ways: (i) by preventing the development of new cancer cells, (ii) by restraining the carcinogens from reaching their activation sites, (iii) by the reduction in toxicity of some compounds by preventing their metabolism [10]. The underlying mechanisms by which flavonoids mediate their function has been well-established for multiple pathways.

Interest in flavonoids production is growing, which is reflected by their 16.5% projected CAGR, increasing the current market from USD 1.9 billion in 2019 to USD 3.5 billion by 2025 [11]. Although the role of flavonoids in the prevention of cancer has been well established, their low availability, issues in isolation and purification of a specific targeted compound, and limited understanding regarding absorption and intestinal metabolism have held back the development of flavonoids as approved drugs for clinical use [10]. It is well known that post-synthesis modifications, such as methylation, glycosylation, phosphorylation and alkylation, all impact pharmacokinetics (PK) and pharmacodynamics (PD). Plants already utilise this chemical space and these modified compounds may be the answer to the challenges facing the clinical usage of flavonoids. These modifications, which are otherwise difficult to access from traditional synthetic chemistry, can be accessed by applying synthetic biology methodologies and metabolic engineering to produce flavonoids in microorganisms.

## 2. Pharmacokinetic Challenges of Flavonoids

Many pharmacological functions are associated with flavonoids; however, a number of problems are holding back their development as approved drugs for clinical use and, to some extent, further research studies (Table 1). Some of them are their low solubility, bioavailability and, to some extent, low yield in host plants, which are discussed in the following sections; however, certain others, like issues in purification from plant sources, and problems in conducting reliable epidemiological studies, have been discussed elsewhere [10].

The absorption and metabolism of flavonoids have been extensively studied in the last two decades. Generally, the PK profile (i.e., absorption, distribution, metabolism, excretion and toxicity) of flavonoids is not optimal and varies considerably across different classes [21]. Flavonoids generally possess low bioavailability when orally administered which significantly decreases their chance of attaining effective concentration in vivo [22]. The reasons for this are their low solubility, poor oral absorption, and extensive hepatic metabolism by phase-I and II enzymes [23].

Intestinal metabolism also affects the absorption of flavonoids via chemical reactions occurring in epithelial cells of the small intestine and/or mediated by the intestinal microbiota [14]. Flavonoids are substrates for glucuronidation, O-methylation and sulfation in small-intestine epithelial cells and these chemical modifications decrease the bioactivity of flavonoids, meaning that these metabolites are excreted [24]. For example, following oral administration in rats, only 20% of quercetin was absorbed in the intestine; the rest was decomposed to CO_2_ as well as excreted in the feces. On the other hand, the absorbed quercetin was also excreted out of the body within 48 h [25]. Quercetin has a low stability profile, as it is degraded within 6 h of incubation under normal physiological conditions (Hanks’ Balanced Salt solution, pH 7.4) [26]. Furthermore, when the unabsorbed flavonoids reach the colon, they undergo degradation into different metabolites by intestinal microflora, mostly via hydrolysis, reduction or ring fission [27,28,29,30].

Surprisingly, the PK profile of flavonoids is also compromised in planta due to environmental factors like light, temperature, oxygen exposure, pH and ultraviolet radiation. Light exposure, specifically UV light, can alter the biosynthesis of flavonoids in host plants. For example, the antioxidant activity of total flavonoids isolated from plant *Halia bara* was optimum at a wavelength of 310 umolm^−2^s^−1^ and any variation in this wavelength results in reduced biosynthesis and bioactivity of flavonoids [31]. Temperature also plays an important role in the extraction and shelf-life of flavonoids. For instance, the optimal temperature for the extraction and purification of flavonoids from pericarp tissue of litchi fruit is 45–60 °C, and other temperatures result in significant yield loss and degradation [32]. As environmental factors are difficult to control and manage, it is impossible to predict the yield and biological activity of flavonoids present in plant extracts.

Flavonoids are present in very minute quantities (micro/milligram per kg of plant biomass) in plant hosts. Therefore, continuous extraction from plant sources, on the one hand, strains an already vulnerable agriculture sector and, on the other hand, might result in price hikes due to unstable supply and demand issues [17]. Flavonoids are commonly available as a plant extract that is a mixture of many plant natural products, therefore making it difficult to link a particular pharmacological effect with a specific flavonoid compound [10]. The presence of multiple secondary metabolites in crude mixture also makes it difficult to isolate and identify a compound of interest. Along with this, isolation and purification methods are costly, hazardous and multistage processes that further decrease the final yield [18]. Taking all these factors into account, it becomes very difficult to predict the final yield, and sometimes makes it impossible to maintain a sustainable supply of required compounds on the market [33]. Therefore, the extraction of flavonoids from plant sources is time-consuming, cost-inefficient, gives very low yield and produces much waste.

Chemical stability, PK issues and low availability are significant obstacles to the clinical development of flavonoids as effective chemopreventive therapy, because the required in vivo level is almost impossible to achieve, even with high oral doses [34,35]. However, there is hope, as a few reports have shown that certain substitutions for flavonoid, i.e., methylation, glycosylation, can improve the PK profile of flavonoids. The roles of methylation and glycosylation are further explained in the following paragraphs and, in the next section, we will shed light on synthetic biology approaches that can help to improve/solve the issues of bioavailability and pharmacokinetics faced by flavonoids.

## 3. Flavonoids Derivative with Improved PK Characteristics

The chemical structure of flavonoids generally, and different substitutions of the basic skeleton specifically, define chemical stability and bioavailability. It is now widely accepted that the substitution of basic flavonoid skeleton has a strong influence on the absorption, distribution and metabolism of flavonoids [36,37].

## 4. Methylated Flavonoids

Methylation, the addition of a methyl group to a substrate, controls several important functions of cells, from gene regulation through epigenetics to maintaining cellular energy status [38]. Depending upon the site, methylated flavonoids are divided into two types: O-methylated flavonoids, ones that obtain a methyl group through hydroxyl group and C-methylated flavonoids, in which the methyl group is directly bound to C atoms of the basic skeleton (Figure 2). Both the O-methylation and C-methylation reactions are catalysed by their respective O-methyltransferases (OMT) and C-methyltransferases (CMT). The methyl group is usually donated by an electrophilic *S*-adenosyl-L-methionine (SAM) through a biomolecular nuclear substitution (SN2) reaction [38,39]. Catechol-OMT, a caffeic acid methyltransferase, is the first SAM-dependent methyltransferase to be crystallized and represented a critical milestone in drug research [40].

The role of methylation in improving the metabolic stability of flavonoids has been documented [41]. O-methylated flavonoids have better bioavailability because of its better absorption and increased permeability across membranes [42,43,44]. In one study, over 8–10-fold better intestinal absorption was documented for methylated flavonoids compared with their non-methylated parent molecules [42]. Methylation is considered to be the main reason behind for the absorption and improved stability. The improved metabolic stability is mainly based on the hypothesis that the blocking of flavonoid-free hydroxyl groups can help in reducing the conjugation reaction by glucuronidation and sulfation, the primary factors responsible for the poor bioavailability and stability of flavonoids [45]. The case of galangin and its methylated derivatives is another interesting example in support of this hypothesis, where it was reported that nearly 90% of galangin was metabolised by glucuronidation and sulfation within 1 h of incubation [46]. On the other hand, the methylated derivatives, 3′,4′-dimethoxyflavone, showed great resistance to metabolism and 5,7-dimethoxyflavone remained relatively unaffected, showing the improved metabolic stability of methylated flavonoids [46].

Methylated derivatives of flavonoids usually show higher bioactivity, and the site as well as extent of methylation play an important role. B-ring methylation appears to have greater anti-cancer potential, where a study of over 30 different flavonoids showed that methylation of 4-C of the B-ring is linked with higher potency [47]. For example, hesperetin is a stronger inhibitor as compared with the eriodictyol, diosmetin is more active than luteolin, acacetin is a strong inhibitor than apigenin and kaempferide is more potent than kaempferol when tested against breast cancer resistance protein (BCRP) in the *BCRP*-transduced human-leukemia K562 cells [47]. This phenomenon is also observed with methylated forms of chrysin and apigenin, which showed 10 and 8 times more potency against oral squamous carcinoma than their non-methylated parent compounds [48]. Similarly, nobiletin and tangeretin (polymethoxylated isoflavones) have shown higher proliferative inhibition among many Ougan flavonoids [49]. The authors have concluded that 3′-O-methylation is also linked with the enhanced anti-proliferative function of nobiletin. Similarly, a comparatively more methylated flavonoid 3′,4′,7-trimethoxyflavone has a stronger inhibitory effect on BCRP as compared with less methylated acacetin, which is one of the strongest BCRP inhibitor flavonoid [47,50]. These are few examples highlighting methylated flavonoids as more stable, potent and bioavailable chemotherapeutic agents compared to their non-methylated analogs.

## 5. Glycosylated Flavonoids

Glycosylation affects physicochemical properties, immunogenicity, and PK/PD characteristics of chemical compounds [51,52]. The glycosylation of flavonoids to form O-, or C-linked glycosides is viewed as a general route to address issues of poor solubility, stability, and toxicity and, further, is an area of intense research [53,54]. Glycosylated flavonoids are categorised as O-glycosides or C-glycosides based on the type of glycosidic bond with the flavonoid basic skeleton (Figure 3). In case of O-glycosides, the sugar moiety is attached to the basic skeleton via hydroxyl bond (commonly at 3-C and 7-C hydroxyl positions), and, in the case of C-glycoside, the sugar molecule is linked to the flavonoid basic skeleton by their respective carbon atoms (commonly at C-6 and C-8 positions) [55]. Generally, O-glycosylation is common in flavones and flavanols sub-classes and C-glycosylation is common in flavones sub-class [54].

Flavonoid glycosides are generally soluble in water and alcohol; however, a few flavonoids, such as rutin and hesperidin, are sparingly soluble. Conversely, aglycans (non-glycosylated parent flavonoid) usually dissolve in non-polar solvents [56]. As well as this, glycosylation increases the chemical stability of flavonoids in vitro. For instance, Srivastava and Gupta have reported that Chamomile glycosides (dominantly apigenin-7-*O*-glucoside) were highly stable in their solutions under a range of storage conditions (temperature, pH and solvent) [13]. Improvement in stability is a desirable clinical characteristic, and thus glycosylated flavonoids are viewed with great promise.

Anti-BCRP activity has also been observed for a few glycosylated flavonoids, for example, apigenin-7-glucoside and luteolin-4′-*O*-glucoside and the possible reason for this might be their better water-solubility and higher absorption [57]. In another study, it was documented that glycosylated flavonoid (daidzin), when ingested in pure form, has higher systemic bioavailability and plasma concentration compared with non-glycosylated parent flavonoid (daidzein) in healthy men [58]. Thus, glycosylation improves the bioavailability of flavonoids, and helps them to retain their native skeleton, which results in a higher inhibitory effect; therefore, such findings are viewed with great importance [57,58]. On the other hand, aglycans, being insoluble in water, are difficult to administer and, therefore, glycosylated flavonoids are better options.

On the other hand, in some cases, glycosylated flavonoids tend to show more variable bioavailability than their aglycans [59,60]. This was explained by the literature on the flavonoid quercetin, where different glycosides show varied absorption rates and absorption sites, with quercetin-4′-*O*-glucoside having metabolites which are five times more available compared to the metabolites of quercetin-3-*O*-rutinoside [61]. Similarly, not all glycosylated flavonoids have shown higher anti-BCRP activity compared with their aglycans, which indicates that glycosylation has variable effects [57]. Therefore, the basic skeleton of flavonoids, as well as the specific sites of attachment and type of sugar unit, play a role in determining the possible pharmacological outcome and, to keep the effects of glycosylation simple to understand, we have mainly considered glucosylated flavonoids as examples.

## 6. Glycosylation Biosynthetic Pathways

Glycosylation reactions of flavonoids are carried out by enzymes like glycosyltransferases (GT) and glycosylhydrolases (GH), comprising GH13 and GH70 families [62]. At present, there are 114 GT families (Crazy database, February 2021) and, because of their large inventory of receptors and donors, they are preferred to GH. While the utilisation of relatively simple sugars like sucrose makes GH enzymes preferable for simple changes, such as those targeting pharmacokinetics, further understanding and streamlining of donor and receptor data is required for widespread GH applications [62]. A study conducted by Ye et al. led to the characterisation of 11 glycotransferases, which can not only efficiently glycosylate flavonoids, but also chalcones, triterpenoids and licorice compounds, widening the scope of enzymatic synthesis of natural products for various applications [63]. With C-glycosides synthesis, C-glycosyltransferases are an important tool, as synthetic approaches still struggle with the synthesis of bis-C-glycosides, which are the main interest for potential pharmaceutical applications [55].

## 7. Chemical Synthesis of Flavonoids and Flavonoid Derivative

Due to the immense potential of flavonoid, chemical synthesis methods have been explored since the beginning of 20th century, aided by the rapid development of new protocols. In fact, the first synthetic flavonoid glycoside, anthocyanin, was synthesized in 1926 by Robertson and Robinson, which is a 3-O-Glycoside of anthocyanidins [64]. Since then, many subsequent chemical glycosylation methods have developed and they all can be categorized into two different types, based on their approach, as tactic-I and tactic-II. With the tactic-I approach, a sugar moiety is directly attached to the desired flavonoid and tactic-II involves the development of flavonoid moiety after the establishment of glycosidic bonds. Many protocols have been developed since then, such as Koenigs-Knoor, PTC-Protocol or the recent Kondo et al. methods; however, they all have drawbacks, such as low yield, the requirement for hazardous and expensive reagents and constant coupling, decoupling and protective steps, making the synthesis elaborate and expensive [65,66]. Even with all these shortcomings, synthetic approaches have yet to come up with approaches to form complex glycosides and glycoside with unusual bonds.

## 8. Synthetic Biology and Flavonoids

Flavonoid biosynthesis pathways, like other plant secondary metabolites, are very complex and operate under a tight multi-level regulation. The biosynthesis of particular flavonoids depends not only upon pathway-specific enzymes but also on their interaction with other, competing partners; therefore, the genetic engineering of native hosts is difficult [33,67]. Realizing these issues, efforts were made to replicate the biosynthetic pathways of a particular flavonoid into well-characterized hosts such as *Escherichia coli* and *Saccharomyces cerevisiae* [68]. The successful synthesis of artemisinic acid by *E. coli* and *S. cerevisiae* was a huge breakthrough, paving the way to the use of microbial systems as an alternative synthetic platform for the synthesis of plant natural products [69]. With recent developments in synthetic biology and metabolic engineering, and the potential these disciplines have, it is envisaged that the issues which have slowed down flavonoid drug discovery will be addressed in the near future (Figure 4).

In the following, recent developments in synthetic biology and metabolic engineering approaches are discussed, which will directly or indirectly help us to synthesize flavonoids (and derivatives) with an improved PK profile.

## 9. Microbial Systems for Production of Flavonoids

Modern metabolic engineering approaches focus on the (re)construction of metabolic pathways in suitable microbial hosts [70]. This can be done either by importing the complete pathway from the host plant to a microbial chassis or by the introduction of more efficient natural or engineered enzymes, and, in recent years, de novo enzymes have also been added to make a pathway more efficient [71]. The construction of complete in vivo enzyme cascades is desirable because they exclude the need for the addition of costly pathway intermediates, as these cascades rely on hosts’ synthesized biomolecules, cofactors and coenzymes.

Numerous impressive examples of microbial engineering to produce flavonoids in microbes have been reported in recent years (Table 2). The de novo synthesis of naringenin in *S. cerevisiae* [72], and *E. coli* [73] strains separately, or in co-culture [74], synthesis of resokaempferol and fistin by *S. cerevisiae* [75] and the production of naringenin, eriodictyol and taxifolin by *Yarrowia lipolytica* [76] are a few interesting examples. However, there are many challenges associated with linking cellular growth to heterologous product synthesis and, therefore, most of the time, it is difficult to achieve viable titer, rate and yield (TRY) [77]. Expanding on the range of suitable microbial hosts, and co-culturing approaches to manage the load on a single strain are few exciting areas, which are viewed with great promise, regarding their ability to help obtain the industrially acceptable TRY.

**Table 2 biomolecules-11-00754-t002:** Examples of de novo flavonoids biosynthesis in microbes.

Scheme	Compound	Host Organism	Precursors	Titer or Productivity (mg/L)		Approaches	Reference
				Initial	Final		
Flavanones	Pinocembrin	*E. coli*	Glucose	102.0	165.3	Managing precursors balance in prokaryotic cell to achieve highest possible yield	[78]
	Naringenin	*E. coli*	D-glucose	90.59	100.64	Engineering primary metabolism to increase heterologous synthesis of flavonoids	[73]
	Naringenin	*Y. lipolytica*	Xylose	239.1	715.3	Engineering xylose metabolism to increase heterologous synthesis of flavonoids	[79]
	Eriodictyol	*Streptomyces albus*	Sucrose	-	0.002	Exploration of new host for industrial production of flavonoids	[80]
Flavones	Apigenin	*S. albus*	Sucrose	-	0.08	Exploration of new host for industrial production of flavonoids	[80]
	Chrysin	*E. coli*	Phenylalanine	-	9.4	Functional expression of plant enzymes in prokaryotic system	[81]
	Scutellarein	*E. coli*	L-tyrosine	47.1	106.5	Expression of plant P450 enzyme and precursor balancing in prokaryotic system	[82]
Flavonols	Kaempferol	*S. cerevisiae*	Sucrose and glycerol	86	200	Co-culturing for management of metabolic burden and gene expression	[83]
	Quercetin	*S. albus*	Sucrose	-	0.1	De novo synthesis of flavonoids in industrial actinomycetes	[84]
	Galangin	*E. coli*	Phenylalanine	-	1.1	Functional expression of plant enzymes in prokaryotic system	[81]
Isoflavanones	Genistin	*E. coli*	Genistein	-	75.9	Bioconversion of isoflavonoids into their glycosylated forms	[85]
	4′-O-methyl daidzein	*E. coli*	Daidzein	49.4	102.8	Enzyme screening and precursor management for synthesis of flavonoid derivatives	[86]
	4′-O-methyl genistein	*E. coli*	Genistein	25.7	46.8	Enzyme screening and precursor management for synthesis of flavonoid derivatives	[86]

## 10. Cell-Free Metabolic Engineering Approaches for Production of Flavonoids

Synthetic biology efforts are mostly associated with living organisms; however, the rise in cell free systems as a new platform for synthetic biology has been seen in recent decades. Due to its inherent nature, cell-free metabolic engineering (CFME) provides an open reaction environment, can aid metabolic engineering in various ways, and has been adopted by the research community to probe metabolic pathways [87]. Cell-free systems can speed up the design, build, test and learn cycle by helping in the (re)construction of biosynthetic pathways in vitro and offer a number of advantages, such as substrate diffusion across the cell membrane, toxicity issues, and issues associated with the expression of heterologous genes and precise control over reaction conditions [88]. Cell-free systems are very simple in their approach, as lysates (or (semi-)purified proteins) from different hosts (plants, microbes, etc.) are combined in a mix-and-match approach, which makes it possible to introduce or skip any enzyme of the pathway, to obtain any product of interest.

Cell-free metabolic engineering is developing into a powerful approach to produce complex natural product biomolecules, and has successfully been used for the synthesis of flavonoids. Recently, Ying Zhang et al. demonstrated the in vitro biosynthesis of naringenin using a cell-free system, where they were able to produce 11.22 mg/L of product in a three-hour incubation [89]. The reported method required multiple rounds of optimisation and included optimising the enzyme ratio, substrate concentration, co-factor concentration and reaction conditions. Such freedom to adjust the enzyme ratio per their catalytic rate makes CFME attractive for flavonoid biosynthesis because it allows us to balance the optimum concentration of the slowest enzymes (tyrosine ammonia lyase, 4-coumeryl CoA and chalcone synthase) of the flavonoid pathway in accordance with need [76]. Additionally, the ability to fine-tune the reaction environment to control the side products is also an advantage of cell-free systems as the formation of side products in a heterologous host, due to the promiscuous activity of the host’s enzymes, is always an difficult-to-solve issue [90]. Thus, it is easy to trace the overaccumulation or utilization of intermediates and the formation of side products in a CFME system as compared with a living cell.

Cell-free systems offer multiple advantages, and can sometimes can be used along with cell-based systems to synthesize complex molecules [91], optimize metabolic pathways, for molecular sensing and to implement genetic networks [92]. However, cell-free systems are not economically feasible at present as compared with cell-based systems, as the cost of in vitro protein synthesis is very high [93]. Efforts are also underway to further reduce the cost of cell-free reactions and the development of a protocol by Kovtun et al. for high-throughput protein expression, using *Leishmania* cell-free lysate, is a significant progress towards the economic feasibility of CFME [94].

## 11. Cell-Free Glycosylation Approaches

The limited availability and challenges in the synthesis of structurally homogenous glycosylated natural products has restricted our understanding of the glycosylation process as well as its applications in biotechnology. Unlike DNA and protein biosynthesis, glycosylation is not a template-driven process; instead, it is carried by a series of glycosylation reactions catalysed by specific glycosyltransferase (GT) enzymes, localized at different subcellular locations [95]. The glycosylation process is highly complex, a defining factor for cell viability, and is tightly regulated inside a living cell. Small variations in the glycosylation network severely decrease cell fitness, and all these factors further complicate glyco-engineering efforts in living cells [96]. Following the emergence of CFME as a new production platform, methods for the investigation and manipulation of glycosylation of biomolecules out-side the living cell have been developed, leading to a new field known as cell-free synthetic glycobiology [93]. Although still nascent, cell-free synthetic glycobiology is helping to understand the mechanism of glycosylation reactions and has enabled the synthesis of homogeneous glycosylated flavonoids.

Many enzyme cascades and biochemical pathways have been established in the cell-free format for the synthesis of natural products [97]. As mentioned above, clinical evaluation and utility of flavonoids is limited due to their PK issues and the modification of flavonoids with sugar moiety is a universal way of circumventing these limitations [95]. Therefore, Leloir type glycosyltransferases (GTs), along with different types of glycosyl donors, are characterized for glycosylation reactions in vitro. For instance, OleD from *Streptomyces antibiotics* and YjiC from many *Bacillus* species are most commonly used for cell-free glycosylation of small molecules, and these enzymes can accept a diverse set of NDP-sugars as glycosyl donors and have promiscuous substrate specificity [98,99,100]. In a pilot-scale cell-free reaction study, a purified OleD has performed the glycosylation of more than 100 small molecules, including flavonoids and alkaloids [99]. Similarly, Sohng and co-workers have demonstrated the glycosylation of 23 structurally diverse flavonoids (with a high ~80–100% conversion rate) by a purified YjiC of *Bacillus licheniformis* [100]. Many other GTs have been characterized along with OleD and YijC, which are multi-functional GTs capable of synthesizing O-, N-, and S-glycosidic linkages [93,101]. Cell-free synthetic glycobiology is an active area of research focused on the development of GT assembly lines for the synthesis of specific glycan structures and it is helped by protein engineering and chemical approaches and, recently, by synthetic biology and metabolic engineering.

A novel mass spectrometry based high-throughput screening (MS-HTS) technique has been developed for the characterization of enzymes produced through cell-free protein synthesis (CFPS) lysate [102]. The platform, known as glycosylation sequence characterization and optimization by rapid expression and screening (GlycoSCORES), uses *E. coli* CFPS with self-assembled monolayers for matrix-assisted desorption/ionization (SAMDI) mass spectrometer and was used to investigate the enzyme’s substrate specificity using 3480 unique peptides and 13,903 unique reaction conditions, finally revealing the optimal glycosylation sequence [102]. Recently, the system has been extended to the analysis of intact glycoproteins, which will help in the identification and characterization of glycosylation enzymes [103]. This system can help in the characterization of other enzymes involved in the biosynthesis of novel flavonoid derivatives, and can help in future studies.

## 12. Enzyme Engineering Approaches for Flavonoids Derivatives

The selection of appropriate enzymes is an important step for the manipulation or construction of a metabolic pathway in native or heterologous host. The field of enzyme engineering has also made significant progress and enhanced the substrate scope, selectivity and activity and, quite interestingly, enzymes with non-natural activities have also been added to the tool box [104].

Enzyme engineering can help synthetic biology and metabolic engineering in two ways. By making the preset enzymes more efficient, enzyme engineering can help in the microbial biosynthesis of flavonoids and, through creating de novo enzymes, it can help in the synthesis of novel flavonoids. The development of artificial metalloenzymes (ArMs), specifically, P450 class enzymes, is a notable example. Recently, an ArM that contains an iridium porphyrin complex has been assembled in the terpene-producing *E. coli* strain and the synthesis of unnatural terpenoids was achieved [105]. The use of ArM in artificially constructed biosynthetic pathways in a microbial host expressing natural and artificial enzymes is an exciting opportunity to produce new-to-nature products in vivo. On one hand, this approach can be used to substitute the C-H bond with other functional groups by using the engineered P450 enzymes and, on the other hand, a new-to-nature core structure can be synthesized by engineering the central pathway enzymes.

Enzyme compartmentalization, to mimic nature’s systems of colocalization of enzymes in space, is an intensive area of research in protein engineering as it has multiple advantages [106,107,108]. It can help to control the generation of by-products through substrate channeling so that heterologous compounds do not mix with endogenous enzymatic machinery, and is beneficial for toxic or labile molecules [109,110]. However, a few recent studies demonstrated that diffusion is not a limiting factor; therefore, substrate channeling is unlikely to improve conversion rates as expected [111,112]. There is certainly a need for more evidence and in-depth analysis to design experiments so that the real situation can be made clear and compared with others to derive better conclusions.

In vitro prototyping and the rapid optimization of biosynthetic enzymes (iPROBE) is another interesting avenue that can help in the synthesis of novel flavonoid derivatives. There is no need for large-scale DNA assembly or the metabolic engineering of living cells, so many enzymes and enzyme combinations and cofactor requirements and optimal conditions can be tested in a short period of time [113]. It can also help to find a synergy between the enzyme sets in the context of a full biosynthetic pathway. Therefore, it becomes very easy to create multiple cell-free biosynthesis units that can be assembled in a mix-and-match fashion and many pathways can be analysed.

## 13. Machine Learning

An exciting avenue for synthetic biology and metabolic engineering is the implementation of machine learning algorithms to help understand the metabolic and regulatory networks of host organisms to optimise existing pathways and develop new synthetic routes [114,115].

For modeling gene regulatory networks (GRN), the convolutional neural network (CNN) is a popular machine learning method due to the availability of relevant datasets, reliable prediction and excellent performance in learning unique features linked with biological sequences [116]. Recently, “Convolutional Neural Network for Coexpression (CNNC)” was developed for single-cell expression data analysis as well as to predict sub-cellular interactions and relationships [117]. Similarly, another CNN-based method “Gene regulatory interaction prediction via Deep Learning (GripDL)” was developed for the analysis of spatial expression patterns [118]. CNN also helps in reliable gene annotation, for example, DeepRibo, a CNN and recurrent neural-network-based tool, trained with ribosome binding-patterns and profiling-information, can reliably predict gene-annotation in prokaryotes [119]. Similarly, DeepEC uses three independent CNNs and can reliably predict the enzyme commission number using protein sequence as an input [120]. The availability of accurate information about the GRN and gene annotations can directly help synthetic biology and metabolic engineering in the selection of appropriate host and efficient enzymes.

The reconstruction of a metabolic pathway in a heterologous host is always a difficult task, and is therefore aided by machine learning approaches [121,122]. Recently, a method that integrates three different neural networks with a Monte Carlo tree search algorithm (3N-MCTS) has been developed, which can help in the identification of synthetic routes for the synthesis of a specific target chemical from simple precursors [123]. Following this, machine learning can also help to explore protein sequence variants, not only for the selection of the most efficient enzymes but also in protein engineering, specifically directed evolution approaches for the creation of enzymes with required properties [124]. Following this, machine learning can also help in fine-tuning the gene expression and flux optimization, specifically through genetic modifications using the CRISPR/Cas system. The DeepCRISPR and sgRNA scorer for the prediction of off-target effects and activity of sgRNA, respectively, are especially aided by machine learning [125,126]. In this way, machine learning helps synthetic biology in bottom-up and top-down metabolic engineering approaches for designing efficient microbial systems for the synthesis of any product of interest.

## 14. Conclusions

Flavonoids are an important natural repository for drug discovery, but ready access is limited from the low yields in plants, and their use has been further constrained by stability, bioavailability, and pharmacokinetics in vivo. Synthetic biology and metabolic engineering offer solutions to both the low availability of flavonoids in plants, and their challenging in vivo pharmacokinetic properties. Using these methods, the synthesis of flavonoids and their methylated and/or glycosylated forms can be transferred into ex planta production, either using microorganisms or using cell-free biosynthesis. Additionally, the application of protein engineering techniques and machine-learning algorithms can also provide greater insights into ex planta flavonoid production and yield. A number of flavonoids were successfully produced using microbial hosts in recent years, demonstrating the utility of the approach.

Key to the future of flavonoids in the clinical setting is the understanding and optimisation of their properties for improved efficacy, stability, bioavailability and pharmacokinetics. Recent advances in genome sequencing, coupled with bioinformatics, have provided ready access to data for the identification of endogenous plant methylation and glycosylation enzyme genes, which can then be integrated into ex planta flavonoid biosynthesis pathways. It is this integration of different flavonoid modification enzymes that it is envisaged will open the landscape for the optimisation of flavonoid biophysical and pharmacokinetic properties.

The possible applications of synthetic biology are exciting and almost limitless, but to achieve this, greater insight into how microbial and cell-free systems function is required. Together with this, efforts to use the cheaper and more sustainable substrates such as agricultural leftovers and waste streams will significantly reduce the overall cost as well as potentially increase the productivity. The successful synthesis of flavonoids in a sufficient quantity for research and commercial space will prove groundbreaking for the development of flavonoids as effective drugs.

## Figures and Tables

**Figure 1 biomolecules-11-00754-f001:**
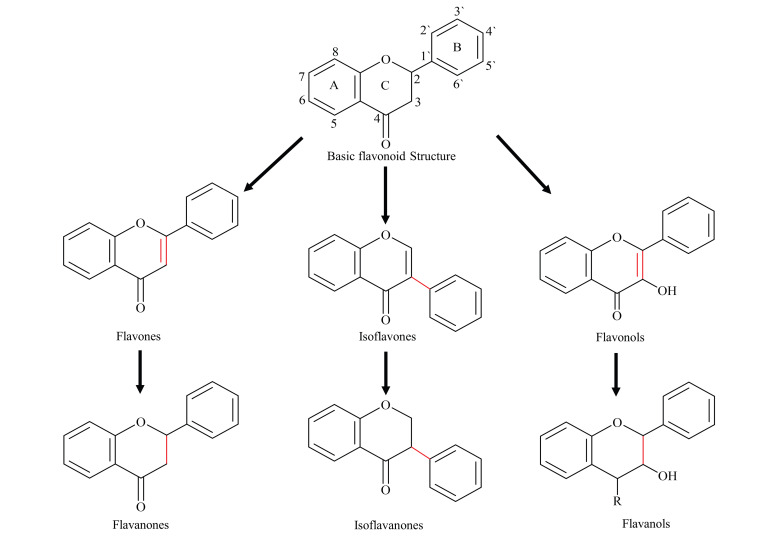
Basic flavonoid backbone and structure of most common sub-classes.

**Figure 2 biomolecules-11-00754-f002:**
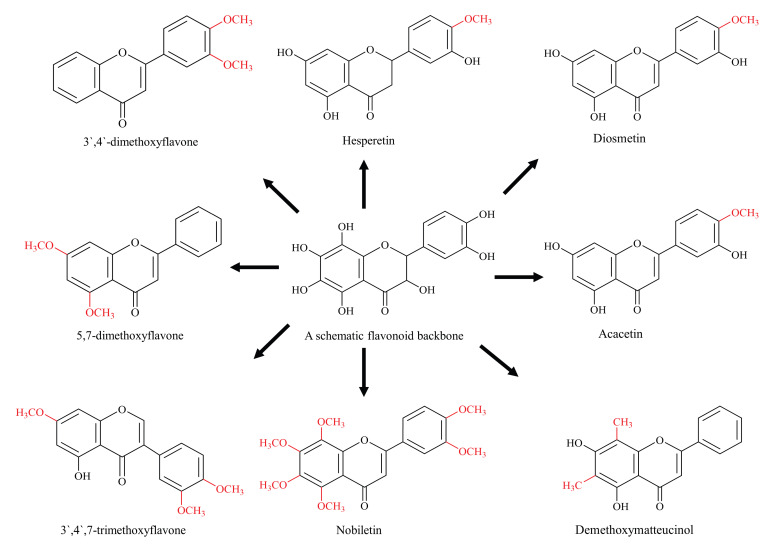
Diagrammatic representation of methylated flavonoids (schematic flavonoid is a hypothetical compound used to show all hydroxyl positions accessible for methylation).

**Figure 3 biomolecules-11-00754-f003:**
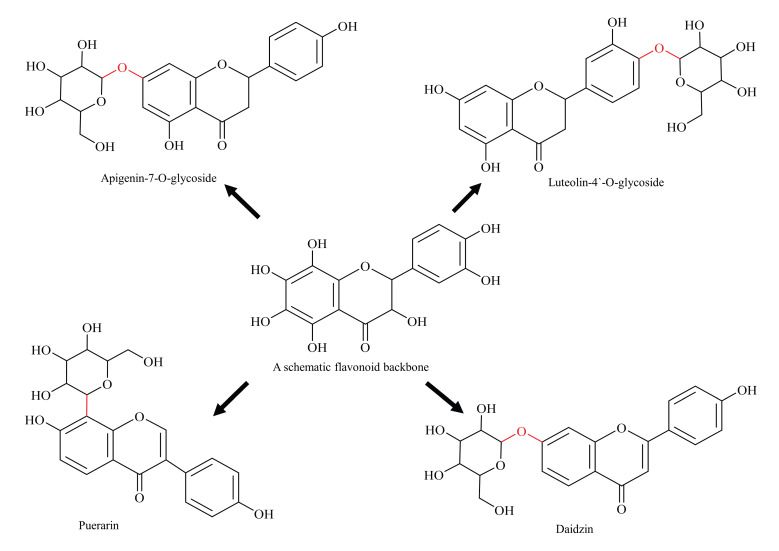
Diagrammatic representation of glycosylated flavonoids (schematic flavonoid is a hypothetical compound used to show all hydroxyl positions accessible for glycosylation).

**Figure 4 biomolecules-11-00754-f004:**
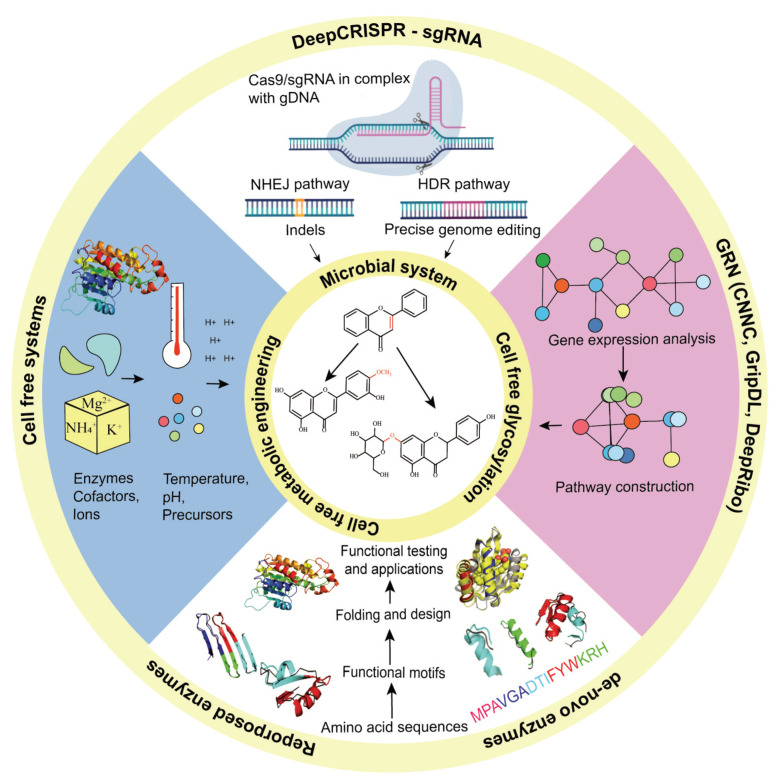
Microbial systems, cell-free systems and cell-free glycosylation approaches are used to synthesize flavonoids with an improved PK profile. To produce commercially viable titer, rate and yield (TRY), machine-learning approaches like DeepCRISPR and sgRNA can help in the genetic engineering of microbial systems and gene regulatory network (GRN) analysis tools can help in the construction of new metabolic pathways. Similarly, protein engineering approaches can also help flavonoid biosynthesis through reproposing enzymes for better activity and stability, as well as by de novo synthesis of enzymes for the production of new-to-nature flavonoid derivatives. Abbreviations: CNNC; convolutional neural network for coexpression, GripDL; gene regulatory interaction prediction via deep learning, NHEJ; non-homologous end joining, HDR; homology directed recombination.

**Table 1 biomolecules-11-00754-t001:** Chemical and biophysical properties, and challenges of flavonoids in planta production. Table 2. Examples of de novo flavonoids biosynthesis in microbes.

Properties	Flavonoid Characteristics	
Solubility	Low intestinal absorption making it difficult to attain pharmacologically effective concentration in-vivo	[12]
Chemical stability	Difficulties in extraction and long-term storage	[13]
Metabolic stability Hepatic, intestinal Intestinal Microflora	Different substitutions on basic skeleton results in lower activity, and inertness which finally leads to excretion Intestinal microflora also results in flavonoids degradation (by hydrolysis, reduction and ring fission)	[10,14]
***In-planta* production constraints**		
Yield	Very low yield of plant secondary metabolites relative to biomass Agricultural and resource constraints to produce sufficient plant biomass	[15,16]
Purity	Heterogeneous mixtures difficult to assign a particular function to a specific molecule Isolation and identification of a particular compound is difficult	[17,18]
Biosynthesis	Regulatory and bioengineering challenges in genetic engineering to increase yield *in-planta*, Seasonal variations in yield and composition	[19]
Isolation and extraction	Loss in activity due to degradation and alteration in chemical structure Production of too much waste during extraction process	[20]

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
