# Peer review of "Synthetic Biology towards Improved Flavonoid Pharmacokinetics"

_biomolecules, 2021, doi:10.3390/biom11050754_

Round 1

Reviewer 1 Report

The manuscript written by Moon Sajid and coworkers described the comprehended wide-scope review for the recent advances in synthetic biology approaches for the heteroloogus expression of plant-specific flovonoids in microbial biosystems. The authors' overall scheme for the review is likely to be logic, and also be fit to BIOMOLECULES scope. Therefore, these kinds of works will be able to spur the production of flavonoids analogs (or derivatives) with enhanced bioactivity potentials. So, as one of the reviewer for this manuscript, I recommend the authors revise a couple of minor comments described below, and that will be suitable enough to be accepted.

Comments;

1) Line 10: "due" should be revised as "due to"

2) Line 34: In figure 1, there is no chemical structure of "anthocyanins" Delete the word "anthocyanins" in the text

3) Line 54: make a space into a word "$1.9billion", making "$1.9 billion"

4) Line 62: revise "flavonoid use" into "clinical usage of flavonoids"

5) Line 74: don't need to use abbriviation "ADMET", becuase the word appears only one time in the full text

6) Line 79: Fully revise the format of table; don't need to put the symbol "hyphen " as well as "bold circle". make the appropriate line alignments in each table cell for better readability

7) Line 97: revise the superscript of "m2s1" unit within the denominator

8) Line 114: the word "optimal" could be revised instead "sustainable"

9) Line 124: an word "abailability" should be revised into "bioavailability"

10) Line 128: the parentheses written in a phrase "~, (methylation, glycosylation) has ~" must be removed

11) Line 135-136: the authos described, in the text, C-methylation. However, there is no typical example for the C-methylation In figure 2. please revised the text, or add the C-methylated chemical structures within figure 2

12) Line 149, 169, and 177: an word "unmethylated" could be replaced with "non-methylated"

13) Line 156: make a space between numeral and unit " 1 h"

14) Line 157 and 158: don't need to give abbreviation "3',4'-DMF" and "5,7-DMF"

15) Line 206-207: check the grammar of paragraph "~of helps flavonoids have ~"

16) Line 255: revise the typo "[34] ]34,67]bil"

17) Table 2: typo written third row "naringeni"

18) Table 2: in row, don't need to write difference format of titer; mean value "715.3 mg/L" will be OK, without the superflous value for standard deviation 

19) Line 297: delete the comma within a phrase "~ different hosts, (plants, microbes etc) ~", giving "~ different hosts (plants, microbes etc) ~"

20) Line 292: don't need to use abbriviation "DBTL", becuase the word appears only one time in the full text

21) Line 308: "fine-tune" will be all right

22) Line 338: the abbreviation should be "CFPS", instead "CEPS"

23) Line 374: describe the full name for "iPROBE"

Author Response

Reviewer 1

The manuscript written by Moon Sajid and coworkers described the comprehended wide-scope review for the recent advances in synthetic biology approaches for the heteroloogus expression of plant-specific flovonoids in microbial biosystems. The authors' overall scheme for the review is likely to be logic, and also be fit to BIOMOLECULES scope. Therefore, these kinds of works will be able to spur the production of flavonoids analogs (or derivatives) with enhanced bioactivity potentials. So, as one of the reviewers for this manuscript, I recommend the authors revise a couple of minor comments described below, and that will be suitable enough to be accepted.

Comments

1) Line 10: "due" should be revised as "due to"

Response: Revised as suggested.

2) Line 34: In figure 1, there is no chemical structure of "anthocyanins" Delete the word "anthocyanins" in the text

Response: “anthocyanins” word is deleted

3) Line 54: make a space into a word "$1.9billion", making "$1.9 billion"

Response: "$1.9billion" is corrected to "$1.9 billion"

4) Line 62: revise "flavonoid use" into "clinical usage of flavonoids"

Response: "flavonoid use" is revised into "clinical usage of flavonoids"

5) Line 74: don't need to use abbreviation "ADMET", because the word appears only one time in the full text

Response: Word "ADMET" is deleted

6) Line 79: Fully revise the format of table; don't need to put the symbol "hyphen " as well as "bold circle". Make the appropriate line alignments in each table cell for better readability

Response: Table is fully revised. Hyphen and bold circles are removed, text is aligned.

7) Line 97: revise the superscript of "m2s1" unit within the denominator

Response: Units are revised

8) Line 114: the word "optimal" could be revised instead "sustainable"

Response: Revised as suggested.

9) Line 124: an word "availability" should be revised into "bioavailability"

Response: a word "availability" is revised into "bioavailability"

10) Line 128: the parentheses written in a phrase "~, (methylation, glycosylation) has ~" must be removed

Response: phrase "~, (methylation, glycosylation) are removed.

11) Line 135-136: the authos described, in the text, C-methylation. However, there is no typical example for the C-methylation In figure 2. please revised the text, or add the C-methylated chemical structures within figure 2

Response: Example of C-methylated flavonoid has been added/Figure 2 is updated.

12) Line 149, 169, and 177: an word "unmethylated" could be replaced with "non-methylated"

Response: word "unmethylated" is replaced with "non-methylated"

13) Line 156: make a space between numeral and unit "1 h"

Response: a space between numeral and unit "1 h" has been made.

14) Line 157 and 158: don't need to give abbreviation "3',4'-DMF" and "5,7-DMF"

Response: abbreviation "3',4'-DMF" and "5,7-DMF" are removed.

15) Line 206-207: check the grammar of paragraph "~of helps flavonoids have ~"

Response: Sentence is revised as below!

Anti-BCRP activity has also been observed for a few glycosylated flavonoids, for ex-ample, apigenin-7-glucoside and luteolin-4`-O-glucoside and the possible reason might be their better water-solubility and higher absorption [57]. In another study, it was documented that glycosylated flavonoid (daidzin), when ingested in pure form, has higher systemic bioavailability and plasma concentration as compared with non-glycosylated parent flavonoid (daidzein) in healthy men [58]. Thus, glycosylation improves bioavailability of flavonoids, and help them to retain their native skeleton, which results in higher inhibitory effect, therefore, such find-ings are viewed with great importance [57,58]. On the other hand, aglycans, being insolu-ble in water, are difficult to administer and therefore, glycosylated flavonoids are better options.

16) Line 255: revise the typo "[34] ]34,67]bil"

Response: Typo is removed.

17) Table 2: typo written third row "naringeni"

Response: Typo is corrected

18) Table 2: in row, don't need to write difference format of titer; mean value "715.3 mg/L" will be OK, without the superflous value for standard deviation 

Response: Correction is made.

19) Line 297: delete the comma within a phrase "~ different hosts, (plants, microbes etc) ~", giving "~ different hosts (plants, microbes etc) ~"

Response: Correction is made.

20) Line 292: don't need to use abbreviation "DBTL", because the word appears only one time in the full text

Response: Word "DBTL" is removed.

21) Line 308: "fine-tune" will be all right

Response: Correction is made.

22) Line 338: the abbreviation should be "CFPS", instead "CEPS"

Response: "CEPS" is changed to "CFPS"

23) Line 374: describe the full name for "iPROBE"

Response: Full name of iPROBE is added/written.

Reviewer 2 Report

Sajid et al., reviewed the ex planta production of flavonoids and their pharmacokinetic properties as it has many beneficial activities for humans. The plant-based production of flavonoids was obtained in low quantity, so as an alternate option, ex planta synthesis of flavonoids by utilizing the advanced synthetic biology approaches like metabolic engineering and cell-free engineering was reviewed along with the methylation and glycosylation approach that facilitated the clinical application of flavonoids.  Initially, the pharmacokinetics challenges of flavonoids and the site-specific modification by methylation and glycosylation that improved biophysical and pharmacokinetics properties were reviewed in detail. Further, how the synthetic biology and cell-free engineering approach was leveraged to produce flavonoids from the microbes was discussed. Finally, concluded as the optimization of the flavonoid production in microbes by enzyme engineering and machine-learning approach. This study provided a comprehensive view about optimizing flavonoid production and its properties by synthetic biology approach.     

Comments

  1. Please cite proper references for Ref. 33, 34 of Page 4 as the cited references were not related to the introduction content provided.
  2. In Page 7, at line 245, please cite a reference to ‘Kondo et al’
  3. In Page 8, in the microbial systems for the production of flavonoids, the examples of microbial engineering to produce flavonoids were shown in Table 2. Please mention the synthetic biology approaches or concepts adopted in each study to produce the flavonoids and show the yield improved on applying the engineering approach.
  4. Please provide a table for the flavonoids produced by the cell-free metabolic engineering approach. Also, discuss with some examples to show the yield improved on obtaining cell-free engineering approach. For example, Ying Zang et al., produced naringenin in vitro from p-coumaric acid by heterologously expressing 4CL, CHS, and CHI. After optimizing the substrate concentration, enzyme ratio, and cofactor concentration, the naringenin production was reached 11.22 mg/L after 3 h. (Ying Zang et al., J. Agric. Food Chem. 2019, 67, 13430−13436).

  1. In Page 9, in the cell-free glycosylation approach, please explain in detail the general concept and types of cell-free glycosylation and how it was used to produce natural products like flavonoids, etc. Finally, discuss the yield and properties improved on using the cell-free glycosylation.

  1. In Page 10, for the enzyme engineering approach, if possible, please provide a concept figure for the approaches utilized in the enzyme engineering studies that improved the production of natural compounds.

  1. Please delete and change the duplicate reference of Ref 17 seen as Ref 24 as both were cited independently on Page 3.    

  1. If possible, please provide an overall concept figure by combining all the approaches utilized in this study improved flavonoid production and its pharmacokinetics properties.

  1. Typos – arrange the hyphen in Table 1

                 – Page 10, line 338, revise ‘CEPS’ to ‘CFPS’

Author Response

Reviewer 2

Sajid et al., reviewed the ex planta production of flavonoids and their pharmacokinetic properties as it has many beneficial activities for humans. The plant-based production of flavonoids was obtained in low quantity, so as an alternate option, ex planta synthesis of flavonoids by utilizing the advanced synthetic biology approaches like metabolic engineering and cell-free engineering was reviewed along with the methylation and glycosylation approach that facilitated the clinical application of flavonoids.  Initially, the pharmacokinetics challenges of flavonoids and the site-specific modification by methylation and glycosylation that improved biophysical and pharmacokinetics properties were reviewed in detail. Further, how the synthetic biology and cell-free engineering approach was leveraged to produce flavonoids from the microbes was discussed. Finally, concluded as the optimization of the flavonoid production in microbes by enzyme engineering and machine-learning approach. This study provided a comprehensive view about optimizing flavonoid production and its properties by synthetic biology approach.     

Comments

  1. Please cite proper references for Ref. 33, 34 of Page 4 as the cited references were not related to the introduction content provided.

Response: Ref 33 and 34 are replaced as suggested.

  1. In Page 7, at line 245, please cite a reference to ‘Kondo et al’

Response: Kondo et al reference is added.          

  1. In Page 8, in the microbial systems for the production of flavonoids, the examples of microbial engineering to produce flavonoids were shown in Table 2. Please mention the synthetic biology approaches or concepts adopted in each study to produce the flavonoids and show the yield improved on applying the engineering approach.

Response: Required information has been added and table 2 is revised.

  1. Please provide a table for the flavonoids produced by the cell-free metabolic engineering approach. Also, discuss with some examples to show the yield improved on obtaining cell-free engineering approach. For example, Ying Zang et al., produced naringenin in vitro from p-coumaric acid by heterologously expressing 4CL, CHS, and CHI. After optimizing the substrate concentration, enzyme ratio, and cofactor concentration, the naringenin production was reached 11.22 mg/L after 3 h. (Ying Zang et al., Agric. Food Chem. 2019, 67, 13430−13436).

Response: Thank you so much for providing a very useful reference. It is hard to find much

data about CFME used for flavonoid biosynthesis to develop a table, however, data from

the suggested reference (Red colour) has been added in the second paragraph (page 10, also

shown below) under Cell free metabolic engineering heading.

  1. Cell free metabolic engineering approaches for production of flavonoids

Synthetic biology efforts are mostly associated with living organisms, however, the rise of cell free systems as a new platform for synthetic biology has been seen during the last couple of decades. Due to its inherent nature, cell free metabolic engineering (CFME) provides an open reaction environment and can aid metabolic engineering in various ways, and has been adopted by the research community to probe metabolic pathways [87]. Cell free systems can speed up the design, built, test and learn cycle by help-ing in (re) construction of biosynthetic pathways in-vitro and offer a number of advantages like substrate diffusion across the cell membrane, toxicity issues, and issues associated with expression of heterologous genes and precise control over reaction conditions [88]. Cell free systems are very simple in their approach, as lysates (or (semi)purified proteins) from different hosts, (plants, microbes etc) are combined in a mix and match approach, which makes it possible to introduce or skip any enzyme of the pathway, to get any product of interest.

Cell-free metabolic engineering is developing into a powerful approach to produce complex natural product biomolecules and has successfully used for synthesis of flavo-noids. Recently Ying Zhang et al. demonstrated the in-vitro biosynthesis of naringenin using a cell-free system, where they were able to produce 11.22mg/L of product in a three-hour incubation [89]. The reported method required multiple rounds of optimisation and included optimising the enzyme ratio, substrate concentration, co-factor concentration and reaction conditions. Such freedom to adjust enzyme ratio as per their catalytic rate makes CFME attractive for flavonoids biosynthesis because it allows us to balance the op-timum concentration of slowest enzymes (tyrosine ammonia lyase, 4-coumeryl CoA and chalcone synthase) of the flavonoids pathway as per the need [76]. Additionally, the ability to fine -tune the reaction environment to control the side products is also an ad-vantage of cell free systems as formation of side products in heterologous host sometimes due to promiscuous activity of host`s enzymes is always an issue which is difficult to solve [90]. Thus, it is easy to trace back the over accumulation or utilization of intermediate and formation of side products in a CFME system as compared with a living cell.

Cell free systems offer multiple advantages, and sometimes, can be used along with cell-based systems to synthesize complex molecules [91], to optimize metabolic pathways, for molecular sensing and to implement genetic networks [92]. However, cell free systems are not economically feasible at the moment as compared with cell-based systems, as the cost of in-vitro protein synthesis is very high [93]. Efforts are also underway to further re-duce the cost of cell-free reactions and development of a protocol by Kovtun et al., for high-throughput protein expression using Leishmania cell free lysate is a significant pro-gress towards economic feasibility of CFME [94].

  1. In Page 9, in the cell-free glycosylation approach, please explain in detail the general concept and types of cell-free glycosylation and how it was used to produce natural products like flavonoids, etc. Finally, discuss the yield and properties improved on using the cell-free glycosylation.

Response: General concept of cell free glycosylation and further explanation has been added under the said heading (Red colour).

  1. Cell free glycosylation approaches

The limited availability and challenges in synthesis of structurally homogenous glycosylated natural products has restricted our understanding of glycosylation process as well as its applications in biotechnology. Unlike DNA and protein biosynthesis, glycosylation is not a template driven process, rather it is carried by a series of glycosylation reactions catalysed by specific glycosyltransferase (GT) enzymes localized at different subcellular locations [95]. The glycosylation process is highly complex, a defining factor for cell viability, and is tightly regulated inside a living cell. Small variations in glycosylation network severely decrease cell fitness and all these factors further complicate glyco-engineering efforts in living cells [96]. Following the emergence of CFME as a new production platform, methods for investigation and manipulation of glycosylation of biomolecules out-side the living cell have been developed leading to a new field known as cell free synthetic glycobiology [93]. Although still nascent, cell free synthetic glycobiology is helping to understand mechanism of glycosylation reactions and has enabled the synthesis of homogeneous glycosylated flavonoids.

Many enzyme cascades and biochemical pathways have been established in cell free format for synthesis of natural products [97]. As mentioned above, clinical evaluation and utility of flavonoids is limited due to their PK issues and modification of flavonoids with sugar moiety is a universal way to circumvent these limitations [95]. Therefore, Leloir type glycosyltransferases (GTs) along with different types of glycosyl donors are characterized for glycosylation reactions in-vitro. For instance, OleD from Streptomyces antibiotics and YjiC from many Bacillus species are most commonly used for cell-free glycosylation of small molecules, and these enzymes can accept a diverse set of NDP-sugars as glycosyl donors and have promiscuous substrate specificity [98-100]. In a pilot-scale cell free reactions study, a purified OleD has performed glycosylation of more than 100 small molecules, including flavonoids and alkaloids [99]. Similarly, Sohng and co-workers have demonstrated glycosylation of 23 structurally diverse flavonoids (with high ~80-100% conversion rate) by a purified YjiC of Bacillus licheniformis [100]. Many other GTs have been characterized along with OleD and YijC which are multi-functional GTs capable of synthesis of O-, N-, and S-glycosidic linkages [93,101]. Cell free synthetic glycobiology is an active area of research focused to develop GT assembly lines for synthesis of specific glycan structures and it is being helped from protein engineering and chemical approaches and recently by synthetic biology and metabolic engineering.

A novel mass spectrometry based high throughput screening (MS-HTS) technique has been developed for characterization of enzymes produced through cell free protein synthesis (CFPS) lysate [102]. The platform known as glycosylation sequence characterization and optimization by rapid expression and screening (GlycoSCORES) uses E. coli CFPS with self-assembled monolayers for matrix assisted desorption/ionization (SAMDI) mass spectrometer and was used to investigate the enzyme's substrate specificity using 3,480 unique peptides and 13,903 unique reaction conditions, finally revealing the optimal glycosylation sequence [102]. Recently, the system has been extended to the analysis of intact glycoproteins that will help in identification and characterization of glycosylation enzymes [103]. This system can help in characterization of other enzymes involved in biosynthesis of novel flavonoid derivatives and can help in future studies.

  1. In Page 10, for the enzyme engineering approach, if possible, please provide a concept figure for the approaches utilized in the enzyme engineering studies that improved the production of natural compounds.

Response: A new image (Image 4) has been added in the text. In image 4, a general pipeline for enzyme engineering is explained/represented that might serve the purpose.

  1. Please delete and change the duplicate reference of Ref 17 seen as Ref 24 as both were cited independently on Page 3.    

Response: Ref 24 is deleted

  1. If possible, please provide an overall concept figure by combining all the approaches utilized in this study improved flavonoid production and its pharmacokinetics properties.

Response: A new image (Image 4) has been added in the text.

  1. Typos – arrange the hyphen in Table 1
    • Page 10, line 338, and revise ‘CEPS ’to CFPS’

Response: Table 1 is revised

- "CEPS" is changed to "CFPS"

Reviewer 3 Report

The authors review the application of synthetic biology approaches to synthesize flavonoids. They first discuss the existing challenges in the synthesis of flavonoids. Next, they discuss recent work in synthetic biology, including cell-free systems and machine learning, which address the challenges.

As a non-expert in the field of flavonoid synthesis, I really enjoy reading the review paper. It concisely presents the challenges and recent solutions. The text flows logically and guides readers to understand various concepts without overusing jargon. I have only one minor comment:

Minor Comment
- The authors should consider adding one figure to summarize the recent synthetic-biology results/approaches (cell-free, machine learning, enzyme engineering) in flavonoid synthesis.

Author Response

Reviewer 3

The authors review the application of synthetic biology approaches to synthesize flavonoids. They first discuss the existing challenges in the synthesis of flavonoids. Next, they discuss recent work in synthetic biology, including cell-free systems and machine learning, which address the challenges. 

As a non-expert in the field of flavonoid synthesis, I really enjoy reading the review paper. It concisely presents the challenges and recent solutions. The text flows logically and guides readers to understand various concepts without overusing jargon. I have only one minor comment:

Minor Comment
- The authors should consider adding one figure to summarize the recent synthetic-biology results/approaches (cell-free, machine learning, enzyme engineering) in flavonoid synthesis.

Response: We thank you for this brilliant suggestion. A new figure (Figure 4) has been added in the text.

Round 2

Reviewer 2 Report

All the comments raised were addressed well by providing the necessary changes in the manuscript. In addition, three more comments were included below. On addressing this, the manuscript can be accepted without any revision.

I recommend that this kind of review is necessary to produce the plant-derived compounds with the advancement of the synthetic biology approach. Moreover, the cell-free systems and machine-learning approaches can overcome the challenges and hurdles of producing plant-derived compounds.

I would like to commend the authors on their work and appreciate their efforts in responding to the concerns of the reviewer.

Minor comments

  1. Typos – please revise the term ‘Titter’ to ‘Titer’ in Table 2 of Page 9 & 10.
  2. In Table 2 of Page 10, please revise the cited ref.82, as it is not the appropriate reference for the flavonol compound ‘Galangin’.
  3. In Figure 4, please increase the font size of the words mentioned in each engineering approach and improve the quality and color of the image to make it comfortable for the readers.  

Author Response

Reviewer 2

All the comments raised were addressed well by providing the necessary changes in the manuscript. In addition, three more comments were included below. On addressing this, the manuscript can be accepted without any revision.

I recommend that this kind of review is necessary to produce the plant-derived compounds with the advancement of the synthetic biology approach. Moreover, the cell-free systems and machine-learning approaches can overcome the challenges and hurdles of producing plant-derived compounds.

I would like to commend the authors on their work and appreciate their efforts in responding to the concerns of the reviewer.

Minor comments

  • Typos – please revise the term ‘Titter’ to ‘Titer’ in Table 2 of Page 9 & 10.

Response: typo is corrected.

  • In Table 2 of Page 10, please revise the cited ref.82, as it is not the appropriate reference for the flavonol compound ‘Galangin’.

Response: Reference is updated. Ref 82 is replaced with 81, which is the correct reference for “Galangin” synthesis in E. coli.  

  • In Figure 4, please increase the font size of the words mentioned in each engineering approach and improve the quality and color of the image to make it comfortable for the readers.  

Response: Text size is increased. The quality and color of the image are also improved.

An updated MS file with track changes has been submitted here.
